# A biomimetic nanoreactor for synergistic chemiexcited photodynamic therapy and starvation therapy against tumor metastasis

Zhengze Yu[1], Ping Zhou[1], Wei Pan[1], Na Li[1] & Bo Tang[1]

Photodynamic therapy (PDT) is ineffective against deeply seated metastatic tumors due to poor penetration of the excitation light. Herein, we developed a biomimetic nanoreactor (bio-NR) to achieve synergistic chemiexcited photodynamic-starvation therapy against tumor metastasis. Photosensitizers on the hollow mesoporous silica nanoparticles (HMSNs) are excited by chemical energy in situ of the deep metastatic tumor to generate singlet oxygen ($^1O_2$) for PDT, and glucose oxidase (GOx) catalyzes glucose into hydrogen peroxide ($H_2O_2$). Remarkably, this process not only blocks the nutrient supply for starvation therapy but also provides $H_2O_2$ to synergistically enhance PDT. Cancer cell membrane coating endows the nanoparticle with biological properties of homologous adhesion and immune escape. Thus, bio-NRs can effectively convert the glucose into $^1O_2$ in metastatic tumors. The excellent therapeutic effects of bio-NRs in vitro and in vivo indicate their great potential for cancer metastasis therapy.

[1] College of Chemistry, Chemical Engineering and Materials Science, Key Laboratory of Molecular and Nano Probes, Ministry of Education, Collaborative Innovation Center of Functionalized Probes for Chemical Imaging in Universities of Shandong, Institute of Molecular and Nano Science, Shandong Normal University, Jinan 250014, China. Correspondence and requests for materials should be addressed to N.L. (email: lina@sdnu.edu.cn) or to B.T. (email: tangb@sdnu.edu.cn)

The metastatic spread of cancer cells is disastrous for patients and often leads to death[1,2]. As a promising candidate for curing cancer, photodynamic therapy (PDT) has performed well and proven to be effective in many cancers over past decades[3–5]. Recent PDT in the clinic focuses on superficial tumors or lesions that are accessible through endoscopes, such as oral cancer, skin cancer, and esophageal cancer[6]. However, it is frustrating that PDT is almost not useful against cancer metastasis because the low penetration of excitation light makes it impossible to reach deep metastasis sites[7–10]. Although researchers have designed near-infrared light triggered photosensitizers to overcome penetration problems, these photosensitizers still suffer from low efficiency[11–13]. To improve the clinical application of PDT to deeply seated metastases, it is feasible to conduct PDT using chemical energy instead of light excitation[14,15]; this chemical energy can be produced by a reaction between hydrogen peroxide ($H_2O_2$) and peroxyoxalate derivatives[16,17]. However, the intracellular $H_2O_2$ concentration is rather low (less than 0.1 μM)[18] and cannot generate sufficient chemical energy, severely limiting chemiluminescence resonance energy transfer (CRET)-based PDT. Cancer starvation therapy is another emerging therapeutic method that blocks nutrient supply to suppress tumor growth[19–21]. Considering the essential role of glucose in cancer cell proliferation and metabolism, we chose glucose oxidase (GOx) to consume intracellular glucose through a glucose-involved reaction that catalyzes the conversion of glucose into gluconic acid and $H_2O_2$[22,23]. Remarkably, this process can not only deplete intracellular glucose for starvation therapy but also increase endogenous $H_2O_2$ levels to generate reactive oxygen species (ROS) for PDT. Thus, chemiexcited PDT combined with starvation therapy is an ideal candidate for treating cancer metastasis.

Furthermore, both the PDT and the oxidation of glucose depend on oxygen ($O_2$). The consumption of $O_2$ will greatly affect the production of $H_2O_2$ and the PDT effect. In addition, the hypoxic properties of the tumor environment, especially those in the inner part of the solid tumor, greatly limit the performance of PDT[24–26]. Consequently, new $O_2$-carrying nanoparticles are expected to enhance the synergistic effects of PDT and starvation therapy. Nanoparticles are traditionally surface functionalized with folic acid, polyethylene glycol, peptides, aptamers, or polymers to improve their tumor-targeting ability[27–31]. However, most of them are still eliminated by the reticuloendothelial system during blood circulation, resulting in low targeting efficiency[32,33]. Cancer cells can perform immune escape and homologous adhesion due to their specific plasma membrane proteins[34–37]. Therefore, biomimetic nanoparticles with cancer cell membranes will greatly improve the delivery efficiency of nanoparticles to tumors.

In the present work, we designed a CRET-based biomimetic nanoreactor (bio-NR) to perform synergistic photodynamic-starvation therapy against tumor metastases by converting glucose into singlet oxygen ($^1O_2$) in cancer cells. Hollow mesoporous silica nanoparticles (HMSNs) are firstly modified with the photosensitizer chlorin e6 (Ce6) and GOx on the surface, followed by co-encapsulating bis[2,4,5-trichloro-6-(pentyloxycarbonyl)phenyl] oxalate (CPPO) and perfluorohexane (PFC) into the cavity of HMSNs, and then coating with cancer cell membrane. Thus, in this bio-NR, Ce6 will be activated by the chemical energy produced from the reaction between CPPO and intracellular $H_2O_2$ to generate ROS via CRET for PDT with no light excitation. At the same time, the conversion of glucose into $H_2O_2$ will be catalyzed by GOx, which not only consumes nutrients for starvation therapy but also enhances PDT synergistically due to the $H_2O_2$ supply. Furthermore, PFC can carry $O_2$ to modulate the hypoxic environment of the tumor and accelerate the rate of glucose oxidation and ROS generation. In addition, the cancer cell membrane coating confers excellent targeting ability via immune escape and homologous adhesion to the nanoreactor. The structure of the bio-NR and the details of using synergetic PDT and starvation therapy via CRET against cancer metastasis are illustrated in Fig. 1.

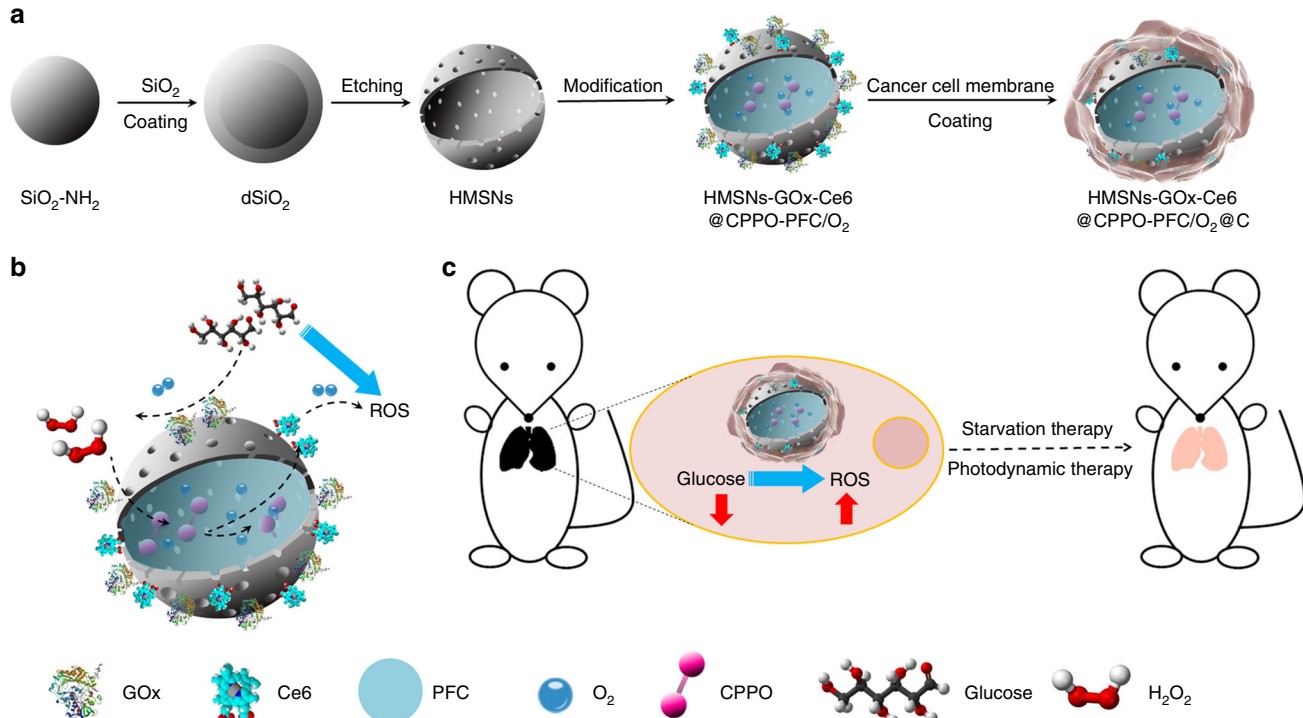

**Fig. 1** Schematic illustrations of the process for synthesizing the biomimetic nanoreactor (**a**), ROS generation based on CRET with glucose consumption with no light excitation (**b**), and synergetic photodynamic-starvation therapy for metastases (**c**)

## Results

**Characterization of the bio-NRs.** HMSNs were synthesized according to a modified method reported previously[38]. In brief, amino-functionalized silica nanoparticles ($SiO_2$-$NH_2$) were first prepared via a reverse microemulsion method. Then, $SiO_2$-$NH_2$ was coated with the $SiO_2$ layer to form a core–shell structure ($SiO_2$-$NH_2$@$SiO_2$, denoted d$SiO_2$). HMSNs were obtained by selectively etching the $SiO_2$ core using hydrofluoric acid (HF), and the mesoporous structure formed simultaneously. For further decoration and cancer cell membrane coating, HMSNs were modified with amino groups. As shown in the transmission electron microscope (TEM) images, $SiO_2$-$NH_2$ and d$SiO_2$ exhibit uniform spherical morphology with diameters of 50 and 80 nm, respectively (Fig. 2a, b). After the $SiO_2$-$NH_2$ core had been selectively etched, HMSNs with a homogeneous shell thickness of approximately 10 nm were present, containing an obvious cavity that can be seen in scanning electron microscope image (Fig. 2c and Supplementary Fig. 1). The modification of amino groups did not destroy the morphology of HMSNs (Fig. 2d). $N_2$ adsorption–desorption isotherms displayed typical Type IV curves and confirmed the mesoporous structure of HMSNs and HMSNs-$NH_2$. The surface area (via the Brunauer–Emmett–Teller (BET) method) and average pore size (via the Barrett, Joyner, and Halenda (BJH) method) of HMSNs were calculated to be 417.17 $m^2$/g and 11.4 nm, and they were decreased to 147.17 $m^2$/g and 8.4 nm after amino modification (Supplementary Fig. 2 and Table 1). The amino content of HMSNs-$NH_2$ was calculated to be 1.12 μmol/mg by thermogravimetric analysis (TGA) (Supplementary Fig. 3). Then, the photosensitizer Ce6 and GOx were anchored on the surface of HMSNs-$NH_2$ via amide reactions, and CPPO and PFC were co-encapsulated into the cavity of HMSNs-$NH_2$ (HMSNs-GOx-Ce6@PFC-CPPO). The characteristic absorption peaks at 400 and 655 nm and the fluorescence emission peak at 665 nm appeared after Ce6 modification, and the Ce6 content was determined to be 0.27 μmol/mg by quantitative fluorescence analysis (Fig. 2f, g and Supplementary Fig. 4). Finally, cancer cell membrane was separated and coated onto the surface of HMSNs-GOx-Ce6@PFC-CPPO to obtain bio-NRs (HMSNs-GOx-Ce6@PFC-CPPO@C). As clearly seen in the TEM image, a lipid layer with a thickness of approximately 10 nm is present, demonstrating the successful coating with the membrane (Fig. 2e). The hydrodynamic diameter of the different nanoparticles was measured using dynamic light scattering; the diameter increased continuously with each step of the process: 108 ± 15 nm for $SiO_2$-$NH_2$, 135 ± 13 nm for d$SiO_2$, 136 ± 11 nm for HMSNs, 143 ± 12 nm for HMSNs-$NH_2$, and 165 ± 11 nm for bio-NRs (Fig. 2h). Moreover, the zeta potential values provided further evidence for the successful construction of nanoparticles in each procedure, with a value of 19.0 ± 1.2 mV for $SiO_2$-$NH_2$, −23.8 ± 1.5 mV for d$SiO_2$, −19.1 ± 0.9 mV for HMSNs, 10.6 ± 0.9 mV for HMSNs-$NH_2$, and −12.9 ± 1.2 mV for bio-NRs (Fig. 2i).

**In vitro verification of the conversion.** The ability of the bio-NRs to convert glucose to $^1O_2$ via CRET was first verified in vitro. In this process, GOx catalyzed the conversion of intracellular glucose into $H_2O_2$, which further reacted with CPPO to produce chemical energy for the photosensitizer Ce6 to generate $^1O_2$. For the detection of $H_2O_2$ produced during the catalytic process, a fluorescence probe that has a specific response to $H_2O_2$, Cy-O-Eb, was employed in fluorescence analysis[39]. As shown in the fluorescence spectra, the intensity of Cy-O-Eb clearly increased when HMSNs-GOx-Ce6 were incubated with glucose (1 mM), indicating $H_2O_2$ generation (Fig. 3a). Furthermore, electron spin resonance (ESR) spectroscopy was also used for the analysis of $H_2O_2$ generation. $H_2O_2$ will convert to hydroxyl radicals (·OH) in the presence of $Fe^{2+}$ through the Fenton reaction, and these radicals can be captured by a radical scavenger, 2,2,6,6-tetra-methylpiperidine (TEMPO)[40,41]. The decreased intensity of the three peaks in the ESR spectra after the addition of glucose demonstrated ·OH generation and further confirmed the production of $H_2O_2$ (Fig. 3b). In addition, the pH value was monitored during the reaction, and it continually decreased (from 7.28 to 3.78 within 60 min) as a result of the production of gluconic acid. These results indicated that GOx remained active after modified on the HMSNs and could catalyze the conversion of glucose into $H_2O_2$ (Fig. 3c). Then, $H_2O_2$ reacts with CPPO in the cavity of HMSNs to form a high-energy intermediate and consequently excite Ce6 to generate $^1O_2$, which was detected using

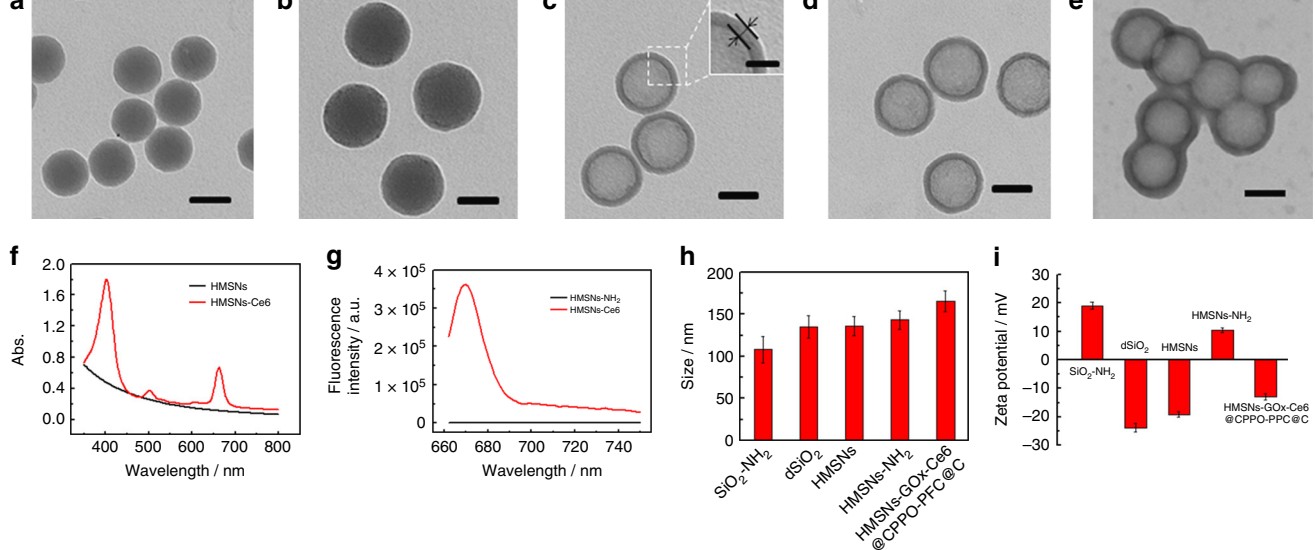

**Fig. 2** Characterization of the biomimetic nanoreactor. TEM images of $SiO_2$-$NH_2$ (**a**), d$SiO_2$ (**b**), HMSNs (**c**), HMSNs-$NH_2$ (**d**), and HMSNs-GOx-Ce6@CPPO-PFC@C (**e**). Scale bar in the inset of (**c**) is 20 nm and others are 50 nm. **f** Absorption spectra of HMSNs and HMSNs-Ce6. **g** Fluorescence spectra of HMSNs and HMSNs-Ce6. Hydrodynamic size distributions (**h**) and zeta potentials (**i**) of the nanoparticles

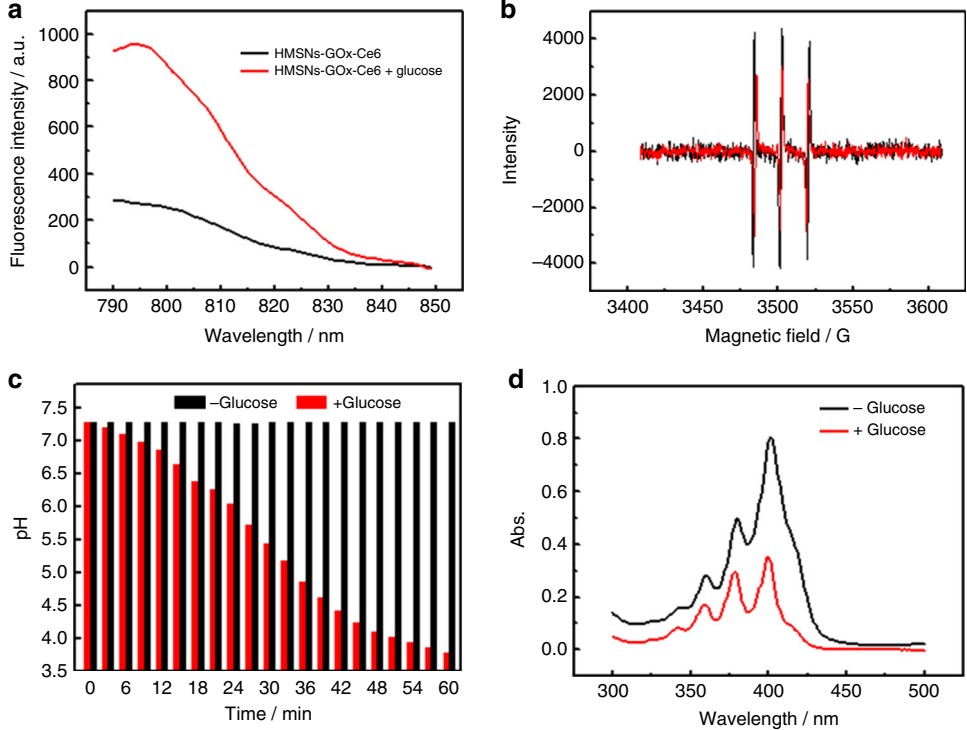

**Fig. 3** Verification of $H_2O_2$ and $^1O_2$ generation in vitro. **a** Fluorescence spectra of Cy-O-Eb probe incubated with HMSNs-GOx-Ce6 in the presence or absence of 1 mM glucose. **b** ESR spectra of TEMPO after incubation with HMSNs-GOx-Ce6 and $Fe^{2+}$ in the presence (red) or absence (black) of glucose. **c** The pH values of HMSNs-GOx-Ce6 solution in the presence or absence of 1 mM glucose. **d** Absorption spectra of ABMD incubated with HMSNs-GOx-Ce6@CPPO-PFC/$O_2$ in the absence or presence of 1 mM glucose

UV–visible (UV–vis) spectrum light by measuring the absorption changes of 9,10-anthracenediyl-bis(methylene)dimalonic acid (ABMD). As shown in Fig. 3d, the absorption intensity decreased sharply when HMSNs-GOx-Ce6@CPPO was incubated with glucose, suggesting the generation of $^1O_2$ from glucose.

**Intracellular therapeutic effects.** Intracellular experiments were carried out to evaluate the bio-NRs in cancer metastasis therapy. Notably, to maximize the simulation of the hypoxic tumor microenvironment, all the experiments were carried out under hypoxic conditions. The ability of the bio-NRs to modulate the hypoxic tumor microenvironment was assessed. Because of the cavity in the HMSNs, bio-NRs can act as a carrier for PFC loading, which can absorb large amounts of $O_2$ due to the van der Waals interactions between PFC and $O_2$. Bio-NRs can thus release $O_2$ in the oxygen-deficient environment. When cells are in the hypoxic environment, they overexpress hypoxia-inducible factor 1α (HIF-1α)[42], which can be detected by immuno-fluorescent staining using confocal laser scanning microscopy (CLSM) in mouse melanoma cells (B16-F10). As shown in Fig. 4a, a bright green fluorescence signal was observed when B16-F10 cells were treated with HMSNs@PFC@C, while the fluorescence intensity was rather low if HMSNs@PFC@C were pretreated with $O_2$ (HMSNs@PFC/$O_2$@C). The results show that HIF-1α level was low in B16-F10 treated with HMSNs@PFC/$O_2$@C, indicating that the nanoparticles carried $O_2$ and modulated the hypoxic tumor microenvironment. Intracellular conversion of glucose into $H_2O_2$ by the bio-NRs in B16-F10 cells was then verified using CLSM. Confocal images show a distinct red signal of the fluorescence probe Cy-O-Eb in B16-F10 cells incubated with HMSNs-GOx@PFC/$O_2$@C or HMSNs-GOx@PFC@C, which demonstrated that bio-NRs catalyzed the conversion of glucose into $H_2O_2$. Moreover, the former materials showed higher

fluorescence intensity than the latter due to the $O_2$ that it carried, indicating a larger amount of $H_2O_2$ generation and revealing $O_2$ to be a key factor (Fig. 4b). We also tested the synergetic photodynamic-starvation therapeutic effect of the bio-NRs against cancer cells with no light excitation via an MTT (3-(4,5-dimethylthiazol-2-yl)-2,5-diphenyltetrazolium bromide) assay. Supplementary Fig. 5 displays the viability of cells receiving different treatments. The data showed that PDT or starvation therapy alone resulted in higher cell viability (73.8% and 62.0%, respectively) than the combination of therapies. Moreover, the viability of cells incubated with bio-NRs lacking $O_2$ reached 57.1%, confirming the important role of $O_2$. Remarkably, the viability of B16-F10 cells was reduced to 20.7% when they were incubated with bio-NRs, indicating their excellent therapeutic effects. What's more, the effect of starvation therapy alone and PDT alone under anaerobic conditions was evaluated and the cell viabilities were 83.7% and 89.6%, which further demonstrated the key role of $O_2$ for both starvation therapy and PDT (Supplementary Fig. 6).

**Homologous adhesion and immune escape in vivo.** As the characteristics of the cancer cell membrane depend on its membrane proteins, we identified five major proteins associated with cell invasion and metastasis (CD44, CD47, E-cadherin, EpCAM, and Tissue factor) using western blot[43–45]. The data showed that these proteins were not destroyed during the extraction of the cell membrane and its coating on the nano-particles (Fig. 4c). Then, the targeting ability of bio-NRs was evaluated via in vivo imaging. B16-F10 cells were injected intra-venously into BalB/C mice to construct a lung metastasis mouse model, which was followed by intravenous injection with HMSNs-Ce6 or HMSNs-Ce6@C. At 24 h post injection, the mice were killed and dissected for in vivo imaging. From in vivo

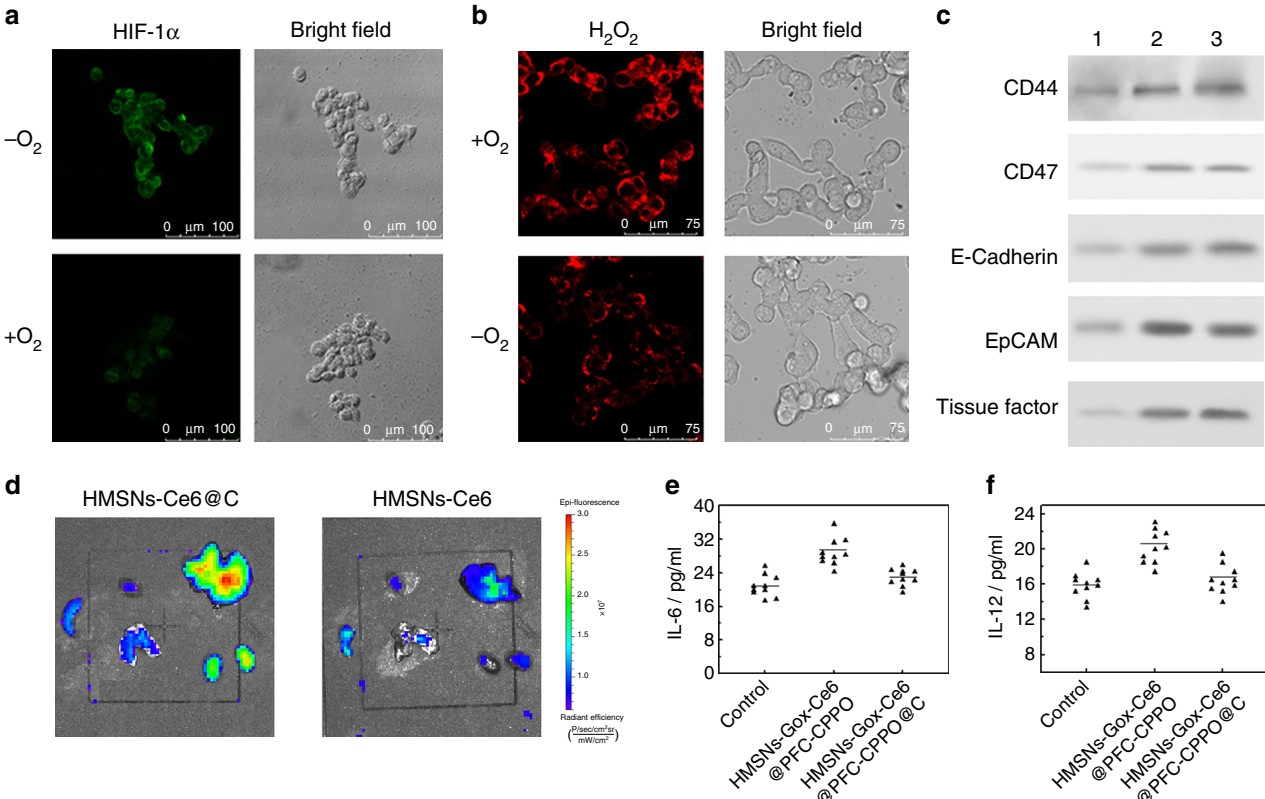

**Fig. 4 a** Immunofluorescent staining images of HIF-1α in B16-F10 pre-incubated with HMSNs@PFC@C (top) or HMSNs@PFC/O$_2$@C (bottom) in a hypoxic environment. **b** CLSM images of B16-F10 cells incubated with Cy-O-Eb and HMSNs-GOx@PFC/O$_2$@C (top) or HMSNs-GOx@PFC@C (bottom). **c** Western blots of CD44, CD47, E-Cadherin, EpCAM, and tissue factor. Column 1, B16-F10 cells; column 2, B16-F10 cell membranes; column 3, HMSNs@C. **d** In vivo imaging of lung with metastatic tumor at 24 h post injection of HMSNs-Ce6@C (left) or HMSNs-Ce6 (right). White arrows point to the lungs. Enzyme-linked immunosorbent assay (ELISA) analysis of IL-6 (**e**) and IL-12 (**f**) after the mice were injected with HMSNs-GOx-Ce6@CPPO-PFC@C or HMSNs-GOx-Ce6@CPPO-PFC

images, we see that lung in the mice treated with HMSNs-Ce6@C exhibited a brighter signal than lung in control mice, indicating the good targeting efficiency after the cancer cell membrane coating (Fig. 4d). Moreover, the fluorescence intensity in the liver and kidney of mice treated with HMSNs-Ce6 was weaker than that of mice treated with HMSNs-Ce6@C, which was mainly due to rapid metabolism by the immune system. These results confirm that nanoparticles can perform homologous adhesion and immune escape. The targeting ability of cancer cell membrane-coated nanoparticles was also assessed using inductively coupled plasma atomic emission spectrometry (ICP-AES) (Supplementary Fig. 7). Data showed that the targeting effect of cell membrane-coated nanoparticles was much better than that of the nanoparticles without the membrane coating. What's more, large amounts of nanoparticles without membrane accumulated in liver and spleen, indicating the rapid clearance. Interleukin-6 (IL-6) and interleukin-12 (IL-12) are secreted by immune cells to stimulate the immune response, and they are often used for the identification of immune activation[46,47]. Therefore, IL-6 and IL-12 were monitored to verify the immune escape of the designed bio-NRs. Figure 4e, f display the expression levels of IL-6 and IL-12 in the blood after mice were injected with bio-NRs or HMSNs-GOx-Ce6@PFC-CPPO. We found that the blood concentration of IL-6 and IL-12 in mice treated with bio-NRs was similar to that of mice without treatment, indicating that bio-NRs indeed achieve immune escape. However, nanoparticles without the membrane coating caused an immune response, reflected by a clear increase in the blood concentration of IL-6 and IL-12; as a

result, these nanoparticles would be cleared from the body faster than the membrane-coated nanoparticles (Supplementary Fig. 8).

**Synergetic therapy for a lung metastatic tumor**. The synergetic photodynamic-starvation therapeutic effect of bio-NRs for metastatic tumors was evaluated in vivo. Figure 5a illustrates the details of the therapeutic process. Mice with lung metastatic tumors were injected intravenously with phosphate-buffered saline (PBS), HMSNs-GOx@CPPO-PFC/O$_2$@C, HMSNs-Ce6@CPPO-PFC/O$_2$@C, HMSNs-GOx-Ce6@CPPO-PFC@C, and HMSNs-GOx-Ce6@CPPO-PFC/O$_2$@C at a dose of 40 mg/kg. After 14 days of treatment, we dissected the lungs and evaluated the therapeutic effect. From the photographs in Fig. 5b, we can see that lung metastatic tumors completely disappeared when the mice received the bio-NRs, demonstrating an excellent therapeutic effect. However, in the control groups, distinct B16-F10 metastatic tumors were observed, and the metastatic rate was rather high (Fig. 5c). Furthermore, the body weight of mice treated with the bio-NRs did not decrease with time, while there were different degrees of weight loss in the other groups due to rapid growth of the metastatic tumors (Fig. 5d). In addition, bio-NR-treated mice exhibited a survival rate of 100%, which further confirmed the high therapeutic efficiency of bio-NRs (Fig. 5e). In addition, hematoxylin and eosin (H&E) staining was employed to verify the synergetic photodynamic-starvation therapeutic effect of bio-NRs. As shown in Fig. 5f, no appreciable lung metastatic

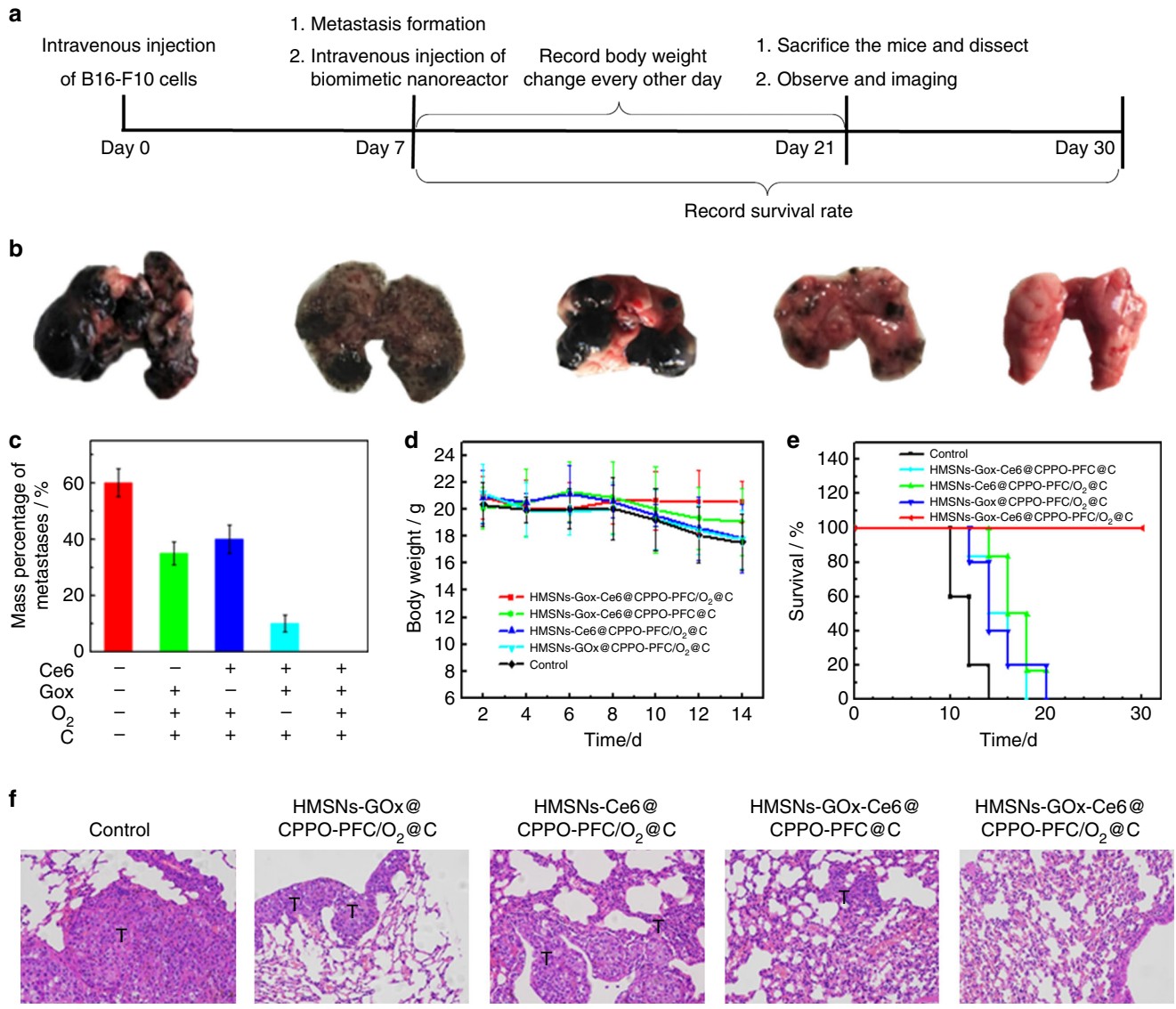

**Fig. 5** Therapeutic effects of the bio-NRs in vivo. **a** Schematic illustration of the in vivo therapeutic process. **b** Macroscopic images of lungs receiving different treatments. From left to right: control, HMSNs-GOx@CPPO-PFC/O$_2$@C, HMSNs-Ce6@CPPO-PFC/O$_2$@C, HMSNs-GOx-Ce6@CPPO-PFC@C, and HMSNs-GOx-Ce6@CPPO-PFC/O$_2$@C. **c** The mass percentage of metastases. Mass percentage of metastases/% = weight of metastatic tumors/weight of normal lung. **d** Body weight curves of mice with metastatic tumors in each group. **e** Survival rates for each group after receiving treatments. **f** H&E staining of lung-bearing metastatic tumors after different treatments (×200, scale bars are 100 μm). T= tumor

tissues were noticed in mice with bio-NRs, indicating the complete removal of metastases. In contrast, the metastatic tumors still existed in all the other groups. The histological effect of the nanoparticles on four major organs (liver, spleen, kidney, and heart) was monitored. H&E stain images showed that there is mild liver injury in mice with nanoparticles that lack the cancer cell membrane coating. This injury occurs because the liver is an immune organ that responds to gather and clear foreign nanoparticles. Nanoparticles without the cancer cell membrane coating cause an immune response and accumulate in the liver. However, cancer cell membrane-coated nanoparticles had no influence on the liver and other organs, suggesting their good biocompatibility (Supplementary Fig. 9).

## Discussion

For the treatment of deep metastatic tumors, an efficient therapeutic method was developed using a bio-NR that conducts synergetic chemiexcited PDT and starvation therapy. To make this nanoreactor, HMSNs were surface-modified with Ce6 and GOx and encapsulated with CPPO in the cavity. Thus, Ce6 could be activated by chemical energy produced from the reaction between CPPO and intracellular H$_2$O$_2$ via CRET, achieving PDT with no light excitation. Remarkably, GOx can catalyze the conversion of intracellular glucose for synergetic starvation therapy, and a reaction product, H$_2$O$_2$, will further react with CPPO to enhance the PDT. Moreover, PFC was co-loaded in the cavity of HMSNs for O$_2$ supply, which solved the problem of O$_2$ lacking and contributed greatly to glucose oxidation and ROS generation. Disguising the nanoparticles with the cancer cell membrane effectively promoted the targeting ability and delivery efficiency of the nanoparticles. The experimental results showed that the PDT therapeutic effect for cancer cells and tumor metastasis was greatly improved after PDT was combined with starvation therapy, indicating that the designed bio-NR has great potential in clinical applications to cancer metastases. We

anticipate that this approach will provide new sight and reference value for clinical cancer metastasis therapy.

## Methods

**Materials and reagents**. 1-Hexanol, cyclohexane, Triton X-100, tetraethyl orthosilicate (TEOS), HF (40%), tetrahydrofuran (THF), 1-(3-diaminopropyl)-3-ethylcarbodiimide hydrochloride (EDC), and N-hydroxysuccinimide (NHS) were purchased from Alfa Aesar Chemical Ltd. (Tianjin, China); (3-aminopropyl)-triethoxysilane (APTES) and CPPO were purchased from Heowns Biochemical Technology Co., Ltd.; Ce6 was purchased from Frontier Scientific Co, Ltd. USA.; TEMPO, protease inhibitors, PFCs, and GOx were purchased from Sigma-Aldrich. Mouse skin melanoma (B16-F10) was purchased from Shanghai Aoluo Bio-technology Co., Ltd. The water used was Mill-Q secondary ultrapure water (18.2 $M\Omega/cm$). The chemical reagents used in the experiment were of analytical grade and used without purification.

**Synthesis of SiO$_2$-NH$_2$**. SiO$_2$-NH$_2$ nanoparticles were prepared by a reverse-phase microemulsion method with some modifications. First, 5.3 mL of Triton X-100, 22.5 mL of cyclohexane, 5.4 mL of n-hexanol, and 1 mL of water were added to a round bottom flask and followed with 750 μL of ammonium hydroxide. After 30 min of stirring, 0.5 mL of TEOS and 0.1 mL of APTES were added to the above solution under vigorous stirring, and the reaction proceeded for 12 h. SiO$_2$-NH$_2$ was obtained by centrifugation (10,000 rpm, 10 min) and washed with absolute ethyl alcohol three times.

**Synthesis of dSiO$_2$**. The as-synthesized SiO$_2$-NH$_2$ was dispersed in 60 mL of absolute ethyl alcohol. Then, 10 mL of water and 10 mL of ammonium hydroxide were added, and the solution was stirred for 30 min. Subsequently, a solution containing 0.3 mL of TEOS and 9.7 mL of absolute ethyl alcohol was dropwise added, and the reaction proceeded for 3 h under stirring. The precipitate, dSiO$_2$, was centrifuged (10,000 rpm, 10 min) and washed with absolute ethyl alcohol and water three times. Finally, dSiO$_2$ was dispersed in 80 mL of water for further use.

**Synthesis of HMSNs**. HMSNs were prepared using a HF etching method. Then, 150 μL of HF solution (4%, w/w) was added to 10 mL of as-synthesized dSiO$_2$. After vigorous stirring for 6 min, the solution was centrifuged immediately (10,000 rpm, 10 min), and the precipitate was washed with absolute ethyl alcohol and water twice. The obtained HMSNs were dispersed in absolute ethyl alcohol.

**Synthesis of HMSNs-NH$_2$**. The HMSNs prepared above were dissolved in 40 mL of anhydrous ethanol, and then 16 mL of water and 400 μL of ammonia were added. After mixing, 10 μL of APTES was added to the above solution. After stirring overnight at room temperature, the mixture was centrifuged (10,000 rpm, 10 min), and the precipitate was washed twice with ethanol and water. Finally, HMSNs-NH$_2$ was dispersed in 10 mL of PBS (pH = 7.4, 0.01 M). The amino groups in HMSNs-NH$_2$ were quantified by TGA.

**Synthesis of HMSNs-GOx-Ce6**. HMSNs-Gox-Ce6 was obtained through the amide-forming reaction between the carboxyl groups of GOx and Ce6 and the amino groups of HMSNs-NH$_2$. EDC (95 mg) and NHS (57 mg) were mixed with GOx (4 mg), and EDC (19.7 mg) and NHS (11.5 mg) were mixed with Ce6 (12 mg) in the dark for 30 min to activate carboxyl groups. The activated GOx and Ce6 were then added to the solution of HMSNs-NH$_2$ under gentle stirring for 24 h. The product HMSNs-GOx-Ce6 were obtained by centrifugation (10,000 rpm, 10 min) and washed three times with PBS buffer to remove unreacted GOx and Ce6.

**Synthesis of HMSNs-GOx-Ce6@CPPO-PFC/O$_2$**. A solution containing 30 mg of HMSNs-Gox-Ce6 was added to a single-necked round-bottomed flask, and the solvent was evaporated by vacuum pump. Then, 1 mL of THF solution containing 15 mg of CPPO was added, and the mixture was sonicated for 2 min to allow the CPPO to enter the HMSN cavity. Subsequently, THF was removed by vacuum to obtain HMSNs-Gox-Ce6@CPPO. Then, 300 μL of perfluorohexane (PFC) solution was added to the powder, and the mixture was sonicated in ice water for 1 min. The redundant solvent was evaporated by vacuum. Finally, the materials were dissolved in a suitable amount of PBS (pH = 7.4, 0.01 M). HMSNs-Gox-Ce6@CPPO-PFC that had been dispersed in PBS were stored in an oxygen chamber (O$_2$ flow rate = 5 L/min) for 10 min to achieve oxygen saturation (HMSNs-GOx-Ce6@CPPO-PFC/O$_2$).

**Cell culture**. The B16-F10 cells used in the experiments were treated with high-glucose Dulbecco's modified Eagle's medium containing 10% fetal bovine serum and 1% 100 U/mL penicillin/streptomycin. Cells were incubated at 37 °C in a humidified atmosphere with 5% CO$_2$. Anaerobic culture conditions were 5% CO$_2$, 1% O$_2$, and 94% N$_2$ at 37 °C.

**Synthesis of the bio-NRs**. B16-F10 cells were first suspended and centrifuged (1000 rpm, 3 min). The cells were washed twice with Tris buffer (pH = 7.4) and then resuspended in Tris buffer with 1% protease inhibitor. Subsequently, the cells were disrupted with a homogenizer in an ice-water bath. Membrane fragments were obtained by differential centrifugation. Membrane fragments were then mixed with HMSNs-Gox-Ce6@CPPO-PFC in an ice-water bath, and the solution was stirred for 24 h. After centrifugation (10,000 rpm, 10 min), the precipitate was separated and redispersed in PBS buffer.

**Verification of the generation of H$_2$O$_2$ and $^1$O$_2$ in vitro**. (1) H$_2$O$_2$: A H$_2$O$_2$-specific molecular probe, Cy-O-Eb, was employed for the detection of H$_2$O$_2$ via fluorescence analysis. Glucose (1 mg/mL) was added to a HMSNs-GOx-Ce6 (2 mg/mL) solution containing Cy-O-Eb (10 μM) at 37 °C, which was then incubated for 12 h. After centrifugation, the fluorescence of the supernate was measured ($\lambda_{ex}$ = 780 nm). H$_2$O$_2$ generation was also verified via ESR by employing the radical scavenger TEMPO. Glucose (1 mg/mL) was added to a HMSNs-GOx-Ce6 (2 mg/mL) solution containing TEMPO and a small amount of Fe$^{2+}$ at 37 °C, which was then incubated for 12 h. After centrifugation, the supernate was analyzed via ESR. Glucose was not added to the control groups. (2) $^1$O$_2$: The generation of $^1$O$_2$ was verified by employing the $^1$O$_2$-specific molecular probe ABMD and analyzing the UV–vis spectra. Glucose (1 mg/mL) was added to the HMSNs-GOx-Ce6 (2 mg/mL) solution containing ABMD (0.1 mM) and then incubated at 37 °C for 12 h, at which point the absorption of ABMD was measured. Glucose was not added to the control groups.

**Intracellular detection of HIF-1α**. In brief, B16-F10 cells were seeded in confocal dishes for 24 h. Then, the cells were incubated with HMSNs@PFC@C or HMSNs@PFC/O$_2$@C (0.2 mg/mL). After further incubation in anaerobic conditions for 4 h, the cells were washed and fixed with precooled 4% paraformaldehyde at room temperature for 20 min; they were then treated with primary antibody and enhanced secondary antibody for 1 h, respectively. Finally, the cells were washed with PBS three times before confocal microscopy experiments ($\lambda_{ex}$ = 488 nm, $\lambda_{em}$ = 500–550 nm).

**Intracellular detection of H$_2$O$_2$**. B16-F10 cells were seeded in confocal dishes for 24 h. Then, the cells were incubated with HMSNs-GOx@PFC@C or HMSNs-GOx@PFC/O$_2$@C (0.2 mg/mL) in anaerobic conditions for 24 h. Cy-O-Eb was added, and the cells were further cultured in anaerobic conditions for 10 min before confocal microscopy experiments. Confocal images were captured with excitation at 633 nm.

**MTT assays**. (1) B16-F10 cells were incubated in 96-well plates and cultured for 24 h under anaerobic conditions. The cells were divided into five groups: control, HMSNs-Ce6@CPPO-PFC/O$_2$@C, HMSNs-GOx@CPPO-PFC/O$_2$@C, HMSNs-GOx-Ce6@CPPO-PFC@C, and HMSNs-GOx-Ce6@CPPO-PFC/O$_2$@C groups. The concentration of the nanoparticles was 0.2 mg/mL. Cells with materials were further incubated in anaerobic conditions for 24 h, and 150 μL of MTT solution (0.5 mg/mL) was then added to each well. After 4 h of treatment, the MTT solution was discarded, and 150 μL of dimethyl sulfoxide (DMSO) was added to dissolve crystals. Finally, the absorbance was measured at 490 nm using a microplate reader (Synergy 2, BioTek, USA). (2) B16-F10 cells were incubated in 96-well plates and cultured for 24 h under anaerobic conditions. The cells were divided into three groups: control, HMSNs-Ce6@CPPO-PFC@C, and HMSNs-GOx@CPPO-PFC@C groups. The concentration of the nanoparticles was 0.2 mg/mL. Cells with materials were further incubated in anaerobic conditions for 24 h, and 150 μL of MTT solution (0.5 mg/mL) was then added to each well. After 4 h of treatment, the MTT solution was discarded, and 150 μL of DMSO was added to dissolve crystals. Finally, the absorbance was measured at 490 nm using a microplate reader (Synergy 2, BioTek, USA).

**Western blot**. Cells were lysed in a radioimmunoprecipitation assay buffer containing 20 mM Tris (pH 7.5), 150 mM NaCl, 50 mM NaF, 1% nonidet P-40, 0.1% deoxycholic acid, 0.1% sodium dodecyl sulfate (SDS), 1 mM EDTA, 1 mM phenylmethylsulfonyl fluoride, and 1 μg/mL leupeptin. Proteins were resolved by SDS–polyacrylamide gel electrophoresis (10%) and transferred onto a polyvinylidene fluoride membrane. The membranes were blocked with 5% fat-free dry milk and incubated with primary antibody overnight at 4 °C. The membranes were incubated with horseradish peroxidase-conjugated secondary antibodies. Specific proteins were visualized with enhanced chemiluminescence detection reagent (Santa Cruz Biotechnology, Inc.). The blots were analyzed using a Bio-Rad imaging system (Bio-Rad, Hercules, CA, USA).

**In vivo targeting and pharmacokinetics by ICP-AES**. All animal experiments were conducted and agreed with the Principles of Laboratory Animal Care (People's Republic of China). BalB/C mice (4–6 weeks old, female, ~20 g) were fed with normal conditions of 12 h light and dark cycles and given access to food and water ad libitum.

BalB/C mice with lung metastatic tumors were injected intravenously with HMSNs-Ce6@C or HMSNs-Ce6 (40 mg/kg). For the targeting experiment, at 24 h post injection, the mice were killed and dissected for imaging. Before the ICP-AES experiment, the metastatic tumors were separated from normal lung tissue by cutting the black B16-F10 metastatic tumors from the total lung tissue. Then, the metastatic tumors and five major organs were dissolved in aqua regia ($HCl:HNO_3:HClO_4 = 3:1:2$, v:v:v) for ICP-AES analysis of Si. For pharmacokinetics, the excrement of the mice was collected at different post-injection times (1, 2, 4, 8, 12, 24, 48, and 72 h) and dissolved in aqua regia ($HCl:HNO_3:HClO_4 = 3:1:2$, v:v:v) for ICP-AES analysis of Si.

**In vivo therapeutic effect of the bio-NRs.** BalB/C mice with lung metastatic tumors were divided into five groups and injected intravenously with PBS, HMSNs-Ce6@CPPO-PFC/$O_2$@C, HMSNs-GOx@CPPO-PFC/$O_2$@C, HMSNs-GOx-Ce6@CPPO-PFC@C, or HMSNs-GOx-Ce6@CPPO-PFC/$O_2$@C. The dose was 40 mg/kg. The mice were killed and dissected at 14 days post injection to assess the therapeutic effect of the nanoparticles. The black B16-F10 metastatic tumors were cut off from the total lung tissue to the maximum extent. Subsequently, the mass of metastatic tumors and the normal lung was then measured using an analytical balance. The mass percentage of metastases was calculated according to the weight of metastatic tumors/weight of normal lung. The body weights of the mice were recorded every other day during 14 days of therapy, and the survival rate was calculated for 30 days. Furthermore, the heart, liver, spleen, lung, and kidney of mice were used for histological sectioning and H&E staining.

## Data availability

All relevant data that support the findings of this study are available from the corresponding author upon reasonable request.

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

## Acknowledgements

This work was supported by the National Natural Science Foundation of China (21535004, 91753111, 21874086, 21775094, 21505087, 21390411) and the Key Research and Development Program of Shandong Province (2018YFJH0502).

## Author contributions

Z.Y., N.L., and B.T. conceived and designed the experiments. Z.Y. and P.Z. performed the experiments. Z.Y., W.P., N.L., and B.T. analyzed the data. Z.Y. and W.P. contributed the schematic materials. Z.Y., N.L., and B.T. co-wrote the paper. Z.Y., W.P., and N.L. edited the manuscript.

## Additional information

**Competing interests:** The authors declare no competing interests.

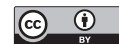

