## [Peer Review File · Nature Communications]

Reviewers' comments:

Reviewer #1 (Remarks to the Author):

The presented work describes a CRET-based biomimetic nanoreactor (bio-NR) to perform synergistic photodynamic-starvation therapy against tumor metastases by converting glucose into 1O₂ in cancer cells. Although this work is interesting, some of the results are contradictory and unconvincing. In addition, some of the following points should also be critically taken into account.

1. The author mentioned that "After amino modification, the surface area and average pore size of HMSNs were decreased from 417.12 m²/g to 147.17 m²/g and 11.4 nm to 8.4 nm, respectively" in the manuscript (page 5 lines 6-9). This data decreased too sharp only with simple amination process. More data should be presented.
2. From in vivo images of Fig. 4d, liver and kidney in the mice treated with HMSNs-Ce6@C exhibited a brighter signal than lung with metastatic tumors. However, Si content in liver and kidney actually lower than that in lung with metastatic tumors (Fig. S6). Moreover, where does the tumor treatment group come from? The mouse model was not lung-bearing metastatic tumors? Please explain it.
3. Fig. 2g&f (Absorption spectra of HMSNs and HMSNs-Ce6 & Fluorescence spectra of HMSNs and HMSNs-Ce6) displayed in the manuscript were actually the TGA & absorption spectra rather than absorption & fluorescence spectra, respectively. Please accurately provide the absorption and fluorescence spectra of HMSNs and HMSNs-Ce6.
4. According to Fig. S5, "The data showed that PDT or starvation therapy alone resulted in higher cell viability (62.0% and 73.8%, respectively) than the combination of therapies" should be corrected as "The data showed that PDT or starvation therapy alone resulted in higher cell viability (73.8% and 62.0%, respectively) than the combination of therapies". At first, the viability of B16F10 cells caused by PDT just reduced to 26.2% after 24 h incubation, this result was not convinced. Moreover, the viability of B16F10 cells treated with starvation therapy has no difference with combination of therapies (in anaerobic conditions) and calculated around 62% (Fig. S5). It means PDT had no effect under anaerobic conditions; this result was too theoretical to be believed.
5. The caption of Fig. 2 and Fig. 4 were confusing: "Hydrodynamic size distributions (f) and zeta potentials (g) of the nanoparticles." should be corrected as "Hydrodynamic size distributions (h) and zeta potentials (i) of the nanoparticles." "ELISA analysis of IL-6 (f) and IL-12 (g) after the mice were injected with HMSNs-GOx-Ce6@CPPO-PFC@C or HMSNs-GOs-Ce6@CPPO-PFC." should be corrected as "ELISA analysis of IL-6 (e) and IL-12 (f) after the mice were injected with HMSNs-GOx-Ce6@CPPO-PFC@C or HMSNs-GOs-Ce6@CPPO-PFC."
6. In Fig. 5F and Fig. S8, the scale bars should be added.
7. CRET should be provided full name when it was first used.
8. In the page 5 lines 9-11 and page 6 lines 8-9, the sentence "The amino content of HMSNs-NH₂ was calculated to be 1.12 μmol/mg by TGA" was repeated twice in this manuscript, please delete one of them.
9. Statistical analysis of the Fig. S5 and S6 should be provided.
10. Authors need to carefully check the language of the paper. There are many obvious errors in the text, such as "Discussion" should be corrected as "Discussion"; In the page 6 lines 7-8, " 1-12.9±1.2 mV for bio-NRs" should be corrected as "-12.9±1.2 mV for bio-NRs". Please check it.

Reviewer #2 (Remarks to the Author):

Authors developed biomimetic nanoreactors which achieve PDT with no light excitation. Furthermore, coating of the nanoreactor with cancer cell membrane increased accumulation to lung by decreasing its clearance in vivo. In the lung metastasis model with B16-F10 melanoma cells, treatment with the biomimetic nanoreactor decreased production of lung metastasis and

prolonged survival of the mice. The effect of the biomimetic nanoreactor seems to be remarkable and its concept is interesting. However, results and methods were not appropriately described. There are apparent mistakes which remarkably decreased the reliability of data as stated below.

Specific comments.

1. Fig.5 (e). Legends indicated that red line shows the control group. However, all mice survived in this group. How all mice in control group could survive? If this is a simple mistake, it remarkably decreases the reliability of results in this paper.

2. Fig.5 (C). This graph may show the percentage of mice with metastasis. If so, the mice inoculated with B16F10 which is known to highly metastatic melanoma cell lines developed lung metastasis in only 60% of mice without therapy. These results are inconsistent with previous reports showing high metastatic potential of this cell line.

3. In Ln 179, authors stated that BalB/C nude mice were used in this experiment. Why nude mice were used for experiments with B16F10 murine melanoma cell line? On the other hand, authors stated in the Methods section (Ln425) that BalB/C mice were used. Authors should carefully explain what type of mice were used for each experiment.

4. How authors determined the dose of biomimetic nanoreactor for in vivo experiments? Authors should state the reason why 40mg/kg biomimetic nanoreactor was used.

5. Ln 225-226. The mean of the sentence "In contrast, many aggressive metastasis can be seen throughout the lung." is unclear.

Minor comment.

1. Ln 222. H&E is not the immunohistochemical staining.

Answers to the questions and changes of the manuscript

Reviewer #1 (Remarks to the Author):

The presented work describes a CRET-based biomimetic nanoreactor (bio-NR) to perform synergistic photodynamic-starvation therapy against tumor metastases by converting glucose into $^1\text{O}_2$ in cancer cells. Although this work is interesting, some of the results are contradictory and unconvincing. In addition, some of the following points should also be critically taken into account.

1. The author mentioned that “After amino modification, the surface area and average pore size of HMSNs were decreased from 417.12 m²/g to 147.17 m²/g and 11.4 nm to 8.4 nm, respectively” in the manuscript (page 5 lines 6-9). This data decreased too sharp only with simple amination process. More data should be presented.

Answer: We thank the reviewer very much. The amination reagent APTES is a silylation reagent, which will react with SiO₂ scaffold and form thin SiO₂ shell on the surface. This process will directly change the pore size, which has a great influence on the surface area of mesoporous materials. And some previous researches based on mesoporous silica reported the similar results (*Angew. Chem. Int. Ed.*, **2010**, *49*, 7281-7283; *Chem. Mater.*, **2012**, *24*, 3895-3905).

2. From *in vivo* images of Fig. 4d, liver and kidney in the mice treated with HMSNs-Ce6@C exhibited a brighter signal than lung with metastatic tumors. However, Si content in liver and kidney actually lower than that in lung with metastatic tumors (Fig. S6). Moreover, where does the tumor treatment group

come from? The mouse model was not lung-bearing metastatic tumors? Please explain it.

Answer: We thank the reviewer very much. We are sorry for the misunderstanding. The mouse model was lung-bearing metastatic tumors. Because the metastatic tumors were separated from the normal lung tissues before Si content measurement by ICP-AES and the mass of metastatic tumors was small, the results showed the calculated ID %/g of Si content in metastatic tumor was higher than that in liver and kidney. And the results robustly demonstrated the good tumor targeting ability of the bio-NRs. The experimental details have been added in the “*In vivo* targeting and pharmacokinetics by ICP-AES.” of methods section.

3. Fig. 2g&f (Absorption spectra of HMSNs and HMSNs-Ce6& Fluorescence spectra of HMSNs and HMSNs-Ce6) displayed in the manuscript were actually the TGA & absorption spectra rather than absorption & fluorescence spectra, respectively. Please accurately provide the absorption and fluorescence spectra of HMSNs and HMSNs-Ce6.

Answer: We thank the reviewer very much for pointing out the mistake. The absorption and fluorescence spectra of HMSNs and HMSNs-Ce6 have been provided in the revised manuscript.

4. According to Fig. S5, “The data showed that PDT or starvation therapy alone resulted in higher cell viability (62.0% and 73.8%, respectively) than the combination of therapies” should be corrected as “The data showed that PDT or

starvation therapy alone resulted in higher cell viability (73.8% and 62.0%, respectively) than the combination of therapies”. At first, the viability of B16F10 cells caused by PDT just reduced to 26.2% after 24 h incubation, this result was not convinced. Moreover, the viability of B16F10 cells treated with starvation therapy has no difference with combination of therapies (in anaerobic conditions) and calculated around 62% (Fig. S5). It means PDT had no effect under anaerobic conditions; this result was too theoretical to be believed.

Answer: We thank the reviewer for the valuable comment. The manuscript has been revised and now the data is consistent with the experiments. Because the intracellular H_2O_2 concentration was very low (less than $0.1 \mu\text{M}$) and it is a key reactant for chemical energy generation, the efficiency of chemiluminescence resonance energy transfer based PDT is much lower than that under light irradiation. In addition, as O_2 is a key factor for both PDT and glucose catalysis in starvation therapy, the anaerobic conditions (1 % O_2) will also severely limit the effect of starvation therapy. We also conducted MTT assay to explain it. The results showed that the cell viabilities of starvation therapy alone and PDT alone under anaerobic conditions were 83.7% and 89.6%, respectively (Supplementary Fig. S6). Therefore, PDT under anaerobic conditions can also have effect on cell death.

5. The caption of Fig. 2 and Fig. 4 were confusing: “Hydrodynamic size distributions (f) and zeta potentials (g) of the nanoparticles.” should be corrected as “Hydrodynamic size distributions (h) and zeta potentials (i) of the

nanoparticles.” “ELISA analysis of IL-6 (f) and IL-12 (g) after the mice were injected with HMSNs-GOx-Ce6@CPPO-PFC@C or HMSNs-GOs-Ce6@CPPO-PFC.” should be corrected as “ELISA analysis of IL-6 (e) and IL-12 (f) after the mice were injected with HMSNs-GOx-Ce6@CPPO-PFC@C or HMSNs-GOs-Ce6@CPPO-PFC.”

Answer: We thank the reviewer very much. We apologize for the carelessness. We have corrected them and check the manuscript carefully.

6. In Fig. 5F and Fig. S8, the scale bars should be added.

Answer: We thank the reviewer very much for the valuable comment. The scale bars have been added in Fig. 5F and Fig. S8.

7. CRET should be provided full name when it was first used.

Answer: We thank the reviewer for bringing this point to our notice. The full name “chemiluminescence resonance energy transfer” has been added in the place where CRET was first used.

8. In the page 5 lines 9-11 and page 6 lines 8-9, the sentence “The amino content of HMSNs-NH₂ was calculated to be 1.12 μmol/mg by TGA” was repeated twice in this manuscript, please delete one of them.

Answer: We thank the reviewer. We have deleted the sentence in the page 6 lines 8-9.

9. Statistical analysis of the Fig. S5 and S6 should be provided.

Answer: We thank the reviewer for the valuable comment. The statistical analysis of cell viability and Si content in different organs was added in the

revised supplementary information.

10. Authors need to carefully check the language of the paper. There are many obvious errors in the text, such as “Discussion” should be corrected as “Discussion”; In the page 6 lines 7-8, ” 1-12.9±1.2 mV for bio-NRs” should be corrected as “-12.9±1.2 mV for bio-NRs”. Please check it.

Answer: We thank the reviewer for pointing out the mistakes. We have revised the manuscript according to the reviewer’s comments and checked the manuscript carefully.

Reviewer #2 (Remarks to the Author):

Authors developed biomimetic nanoreactors which achieve PDT with no light excitation. Furthermore, coating of the nanoreactor with cancer cell membrane increased accumulation to lung by decreasing its clearance in vivo. In the lung metastasis model with B16-F10 melanoma cells, treatment with the biomimetic nanoreactor decreased production of lung metastasis and prolonged survival of the mice. The effect of the biomimetic nanoreactor seems to be remarkable and its concept is interesting. However, results and methods were not appropriately described. There are apparent mistakes which remarkably decreased the reliability of data as stated below.

Specific comments.

1. Fig.5 (e). Legends indicated that red line shows the control group. However, all

mice survived in this group. How all mice in control group could survive? If this is a simple mistake, it remarkably decreases the reliability of results in this paper.

Answer: We thank the reviewer very much for pointing out the mistake. We made an obvious mistake that the control group and HMSNs-GOx-Ce6@CPPO-PFC/O₂@C group were really reverse. We sincerely apologize for our carelessness. We have corrected the mistake in the revised manuscript and checked the manuscript carefully.

2. Fig.5 (C). This graph may show the percentage of mice with metastasis. If so, the mice inoculated with B16F10 which is known to highly metastatic melanoma cell lines developed lung metastasis in only 60% of mice without therapy. These results are inconsistent with previous reports showing high metastatic potential of this cell line.

Answer: We thank the reviewer for the valuable comment. We are so sorry for the misunderstanding. Fig. 5(c) displayed the mass percentage of metastatic tumors in Fig. 5(b). To clear up this confusion, the Y-axis “Metastasis %” was changed to “Mass percentage of metastases”. (Mass percentage of metastases = weight of metastatic tumors / weight of lung). Indeed, almost all the mice inoculated with B16F10 developed lung metastasis in the experiments. The related description has been added in the revised manuscript.

3. In In 179, authors stated that BalB/C nude mice were used in this experiment. Why nude mice were used for experiments with B16F10 murine melanoma cell line? On the other hand, authors stated in the Methods section (ln425) that BalB/C

mice were used. Authors should carefully explain what type of mice were used for each experiment.

Answer: We thank the reviewer very much. BalB/C mice were used in all the animal experiments. We have corrected them in the revised manuscript.

4. How authors determined the dose of biomimetic nanoreactor for in vivo experiments? Authors should state the reason why 40mg/kg biomimetic nanoreactor was used.

Answer: We thank the reviewer very much. According to the previous reports (*ACS Nano*, **2015**, *9*, 2584; *Angew. Chem. Int. Ed.*, **2015**, *54*, 1770; *Angew. Chem. Int. Ed.*, **2018**, *57*, 7759.), the injection dose of nanoparticles for PDT was about 40-100 mg/kg. To minimize the systemic toxicity of the nanoreactors, the dose of 40 mg/kg was chosen for our initial attempt. The data showed that the metastatic tumors could be completely removed with this dose. Therefore, 40 mg/kg biomimetic nanoreactor was used for *in vivo* experiments.

5. Ln 225-226. The mean of the sentence “In contrast, many aggressive metastasis can be seen throughout the lung.” is unclear.

Answer: We are sorry for the confusion. This sentence has been changed to “In contrast, the metastatic tumors still existed in all the other groups.”.

Minor comment.

1. Ln 222. H&E is not the immunohistochemical staining.

Answer: We thank the reviewer. We have corrected it in the revised manuscript.

REVIEWERS' COMMENTS:

Reviewer #1 (Remarks to the Author):

The authors have properly answered all questions with additional data. It is suggested for publication in the present form.

Reviewer #2 (Remarks to the Author):

Comment 2. Authors replied that they showed "Mass percentage of metastases" (Mass percentage of metastases = weight of metastatic tumors/weight of lung. This description is also very unclear, because the methods in detail for measurement of lung and lung metastases for Fig 5(c) was not stated in the Methods section. The red bar in Fig 5 (c) probably shows the mass percentage of metastasis in control group and it is about 60%. However, this reviewer can not understand how authors could measure the weight of metastatic tumors and lung separately. It is also very unclear whether "weight of lung" means weight of total lung with metastases or weight of normal lung.

Response to Reviewer #2

Comment 2. Authors replied that they showed “Mass percentage of metastases” (Mass percentage of metastases = weight of metastatic tumors/weight of lung. This description is also very unclear, because the methods in detail for measurement of lung and lung metastases for Fig 5(c) was not stated in the Methods section. The red bar in Fig 5 (c) probably shows the mass percentage of metastasis in control group and it is about 60%. However, this reviewer can not understand how authors could measure the weight of metastatic tumors and lung separately. It is also very unclear whether “weight of lung” means weight of total lung with metastases or weight of normal lung.

Answer: We thank for the referee very much and sorry for the misunderstanding. The metastatic tumors were separated from normal lung tissue by cutting the black B16-F10 metastatic tumors from the total lung tissue to the maximum extent. Subsequently, the mass of metastatic tumors and normal lung tissue was measured using an analytical balance. And the mass percentage of metastases was calculated according to the weight of metastatic tumors / weight of normal lung. The experimental details have been added in the “*In vivo* therapeutic effect of the bio-NRs” section of Method. In addition, “the weight of lung” means “the weight of normal lung”, which has been changed in the revised manuscript for better understanding.